# EMO-KNOW: A Large Scale Dataset on Emotion and Emotion-cause

**Mia Huong Nguyen♠**     **Yasith Samaradivakara♠♡**     **Prasanth Sasikumar♠**
Suranga Nanayakkara♠     Chitralekha Gupta♠
♠School of Computing, National University of Singapore
♡School of Computing, University of Colombo
{mia, yasith, prasanth, chitra, suranga}@ahlab.org

## Abstract

Emotion-Cause analysis has attracted the attention of researchers in recent years. However, most existing datasets are limited in size and number of emotion categories. They often focus on extracting parts of the document that contain the emotion cause and fail to provide more abstractive, generalizable root cause. To bridge this gap, we introduce a large-scale dataset of emotion causes, derived from 9.8 million cleaned tweets over 15 years. We describe our curation process, which includes a comprehensive pipeline for data gathering, cleaning, labeling, and validation, ensuring the dataset's reliability and richness. We extract emotion labels and provide abstractive summarization of the events causing emotions. The final dataset comprises over 700,000 tweets with corresponding emotion-cause pairs spanning 48 emotion classes, validated by human evaluators. The novelty of our dataset stems from its broad spectrum of emotion classes and the abstractive emotion cause that facilitates the development of an emotion-cause knowledge graph for nuanced reasoning. Our dataset will enable the design of emotion-aware systems that account for the diverse emotional responses of different people for the same event.

## 1 Introduction

Emotion-Cause analysis is gaining interest due to its potential impact on applications such as empathetic dialog systems and mental health support chats. It is more challenging than traditional emotion recognition in text because it requires a higher level of semantic understanding (Lee et al., 2010, Gui et al., 2016, Xia and Ding, 2019).

Most existing works (Gui et al., 2016; Bostan et al., 2020; Kim and Klinger, 2018) model emotion cause analysis as an extraction problem, resulting in datasets that provide only specific, descriptive information on emotion causes and lack deeper, abstract causes. For example, in a tweet, a user expresses their emotion by stating, "I am often on

Tweet: I'm home but it doesn't feel like it. And I'm angry because of it. I hate the feeling of being lonely . They are with me. But I feel empty
Emotion: angry
Cause: feeling lonely and empty despite being home.

Tweet: If you're not making mistakes, you're not doing anything. I feel positive because I make mistakes and I can be better after that.
Emotion: positive
Cause: overcoming mistakes and becoming a better person

Figure 1: Examples of data points in EMO-KNOW

social because I feel very lonely in real life. My friends suck, my relatives suck even more and I have nobody in my life. Even my dog was taken away, so if you are wondering about how I am feeling? Yes, I am lonely." The second sentence is typically annotated as the emotion's cause in existing datasets, but this fails to highlight the more generalized root cause. This way of annotation, however, fails to highlight the more generalized root cause, which in this case is the lack of fulfilling relationships in the person's life. This level of abstraction is essential for comprehensive emotion-cause understanding and reasoning.

A further limitation of existing datasets lies in the coarse granularity of their emotion categories. Modern theories of emotion suggest that emotions are not simply reducible to a basic set of 6-8 emotion categories (Barrett, 2017, Cowen and Keltner, 2017). Therefore, labeling user's emotions with a limited set of emotions will either miss out on many other emotions or forcefully label their emotions as something else. For this reason, a finer granularity of emotion categories would better reflect the true diversity and nuance of human emotions. For instance, 'pride' and 'gratitude' both fall under the category of 'joy,' but each carries distinct nuances and potentially different causes.

While previous work by Zhan et al. (2022) has made progress in summarizing events that trigger emotions, is limited in scale and covers only seven emotions. To enable AI models to reason more

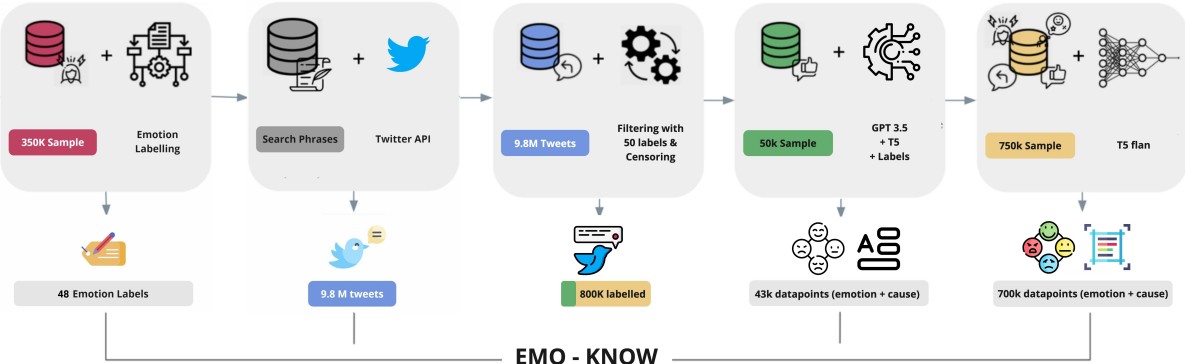

Figure 2: Data generation pipeline

diversely and discerningly about emotion-cause, a comprehensive knowledge base is required. However, manually annotating a large-scale dataset is expensive. In this work, we present EMO-KNOW, a validated dataset on emotion causes curated from 9.8 million tweets over 15 years. Our dataset contains tweets where users describe their emotions and triggering events, spanning 48 emotion types. Unlike other datasets that rely on outside labeling, ours uses the users' own words, making it more genuine. With this data, combined with the emotion-cause details, we can create a map that shows how emotions are linked to causes, helping us understand the connections better. (Wang et al., 2021, Qian et al., 2023). Our contributions are a method to create a large-scale, high ecological validity emotion dataset, validation with automatic metrics and human evaluators, and open-sourcing the dataset[1](700K datums) for researchers.

## 2 Related Work

**Annotated Emotion Cause Datasets.** The field of emotion analysis has seen a rise in interest recently, leading to the creation of several datasets that provide annotations for both emotion and its causes. (Zheng et al., 2022, Chen et al., 2020) For example, Lee et al. (2010) introduced a dataset of 5,964 entries with word-level annotations of emotion and emotion causes, covering 6 emotion classes. Gui et al. (2016) annotated 2,105 news articles with clauses that contain 6 emotion classes and emotion causes. For English corpora, Bostan et al. (2020) annotated a dataset of emotion, emotion cause and affect intensity, comprised of 5000 news headlines spanning 15 emotion classes. Kim and Klinger (2018) introduced a dataset of 1720 triplets

[1]https://github.com/iammia0o/EMO-KNOW.git

of <emotion, emotion experiencer, emotion cause>. Diverging from the extractive approach, Zhan et al. (2022) developed a new dataset focusing on emotion and its triggers by summarizing the events that lead to particular emotions. This dataset has 1883 Reddit posts, annotated with emotions (7 classes) and abstractive summaries of their triggers. As manual annotation often involves the cost of human labor, most annotated datasets are limited in size, and limited in the number of emotion classes they contain.

**Automatic Data Labeling:** In an effort to circumvent the costs of manual annotation, researchers have devised ways to generate large-scale datasets using pre-existing models. For instance, Welivita et al. (2021) produced EDOS by fine-tuning RoBERTa on a small label set created through crowdsourcing, and then used it to label 1 million conversations in the Opensubtitles dataset (Lison et al., 2018). Similarly, Welivita and Pu (2022) created HEAL - a knowledge graph consisting of 1 million distress narratives, annotated for emotion, post summarization, and node clustering using a variety of pre-existing models. A prevalent shortcoming of emotion labeling with current models is their limited accuracy; the emotion classifiers utilized in both HEAL and EDOS only achieved accuracies ranging from 65% to 65.88%. We address this limitation by harnessing users' self-reported emotion labels and rigorously assessing the performance of our fine-tuned emotion-cause summarization model with human evaluators.

## 3 Methods

### 3.1 Data Curation

Data was curated by scraping tweets from Twitter using the Twitter API over a period of 15 years,

spanning from 2008 to 2022. To extract tweets containing emotional expressions and potential emotion causes, we refined search phrases iteratively. We explored the popular emotion expressed by users by examining the frequency of emotion words on a sample of 300K tweets and picked a cut-off at 48. We used this list of 48 emotions to iteratively refine and expand our search space to obtain a set of 9.8 million data. We applied a data cleaning and refinement pipeline, which include removing expression of sympathy, profanity, and tweets with non-meaningful patterns. The process results in a dataset of 772,863 tweets that follows the following patterns:

- `"I X (`$e_1|e_2|..|e_{48}$`) because"`

- `"I X (w+) (`$e_1|e_2|..|e_{48}$`) because"` where X in `{am, feel, am feeling}` and $e_i$ is in the list of 48 emotions.

We detailed our data curation procedure as well as our data statistics in Appendix A.

## 3.2 Data Labeling

### 3.2.1 Emotion Labeling

Since our dataset consists of tweets that follow the outlined structure, we select the emotion words that fit this pattern as the emotion label for each tweet. Although exceptions exist, such as instances of negating statements within tweets, our error analysis found that most cases follow this rule. Thus, we simply extract the emotion label from the word that fits into our identified pattern. This method ensures that the emotion label accurately reflects the individual's true feelings, unlike traditional methods where emotion labels are determined by human annotators, which can introduce biases.

### 3.2.2 Labeling with Large Language Models

| Model | BLEU-2 | BLEU-4 | BERTSc |
|---|---|---|---|
| T5-flan | **0.40** | **0.34** | **0.91** |
| T5-base | 0.36 | 0.31 | 0.89 |
| BART-large | 0.35 | 0.31 | 0.88 |

Table 1: Automatic Evaluation for Candidate Models

After obtaining the emotion label, we proceed to extract the cause of the emotion. We approach this task by treating it as a question-answer challenge. Given a tweet and an emotion label $e$, we train a language model to answer the question: Why

do I feel $e$? At first glance, this task appears simple, especially considering tweets are usually short. Therefore, one might expect traditional Q&A models like BERT (Devlin et al., 2019) to perform well. However, most Q&A models are designed to answer questions in an extractive way, which prevents them from providing abstractive, generalized answers about emotion causes.

Given these limitations of extraction-based models in identifying emotion causes, an alternative approach would be to frame the problem as a generative question-answering problem. This method, however, requires additional training data for pre-trained models. Recent developments in Large Language Models (LLMs) such as GPT-3, InstructGPT (Ouyang et al., 2022) have shown impressive abilities in following instructions and performing tasks like summarization.

LLMs have been trained on a vast amount of text and have often demonstrated a remarkable ability to generalize. Previous studies, like those by Bosselut et al. (2019) have used LLMs to generate abstractive common-sense knowledge relations. These models have also been used to generate pseudo-labels for training smaller models on more specialized tasks, as noted in studies by Ye et al. (2022) and (Taori et al., 2023). This method is very economical. It eliminates the need for human annotators while utilizing freely available pre-trained language models without incurring the full cost of labeling millions of tweets with GPTs. Using this approach, we experimented with various prompts before settling on the following prompt to obtain the emotion and cause labels from InstructGPT: *"Answer the following questions: 1. What is the emotion expressed in the text? 2. What is the cause of such emotion? Provide the answer in the following format strictly: 1. E where E is the emotion from this list: [list of 48 emotions], 2. C where C is the summary of the cause in 10 words. Use NA for E when there is no emotion, and use NA for C when there is no cause."* We selected a random sample of 50,000 tweets to feed into InstructGPT to obtain the emotion-cause labels. After filtering out entries with NA for both emotion and causes, we ended up with 42,956 data points.

Lastly, we evaluated several language model candidates on the training dataset to identify the most suitable model for labeling the entire dataset. For the task of generative text, we considered three candidate models: BART-large, T5-base, and T5-flan.

| Happy | Angry | Sad | Guilty | Proud | Frustrated | Inspired | Alone | Nostalgic |
|---|---|---|---|---|---|---|---|---|
| Friends | Betrayal | Inability | Due | Success | Desire | Actions | People | Missing |
| Life | Inability | Left | Overeating | Achieving | Difficulty | Belief | Connection | Childhood |
| Love | Frustration | Missing | Studying | Completing | Overwhelming | Desire | Despite | Memories |
| Success | Lack | Person | Time | Overcoming | Poor | Influence | Interaction | Time |
| Presence | Loss | Uncertainty | Work | Victory | Understanding | Support | Missing | Playing |

Table 2: Results of topic modeling through LDA(Blei et al., 2003). Keywords are selected from five most prominent topics among abstractive summaries of causes of selected few emotion categories in EMO-KNOW.

These models were trained and assessed on the 42k dataset (which was split into train, val, and test sets in a 6:2:2 ratio). The models' performance was measured using the BLEU scores (Papineni et al., 2002) and BERT-score (Zhang et al., 2020), which are reported in table 1. As shown in the results, T5-flan outperforms all the other models for all the metrics, scoring 0.91 on BERT-score making it very close to the performance of the responses generated by InstructGPT. To guarantee the quality of our dataset, we sample 100 data points and manually inspected the emotion cause generated by the model. We observed that tweets with poor grammar tend to have poor quality of emotion causes. Therefore, we utilize TextAttack's grammar checker [2] model to filter out datapoints with poor grammar. This step leaves the dataset with 698800 tweets. Figure 2 shows a summary of our method and table 2 shows the popular keywords of emotion causes that correspond to a selected number of emotions.

## 4 Evalution

### 4.1 Evaluation criteria

We model emotion-cause identification as an abstractive question-answering task, but the answers are often short summaries. Traditional metrics fail to capture the quality of these summaries (Kryscinski et al., 2020), so we use human evaluation criteria commonly used in summarization tasks. We adapt the criteria to fit the brevity of tweets and use Relevance, Fluency, and Consistency metrics while omitting Coherence, which is typically used for long text. This approach is similar to that used by Zhan et al. (2022). **Relevance** assesses whether the emotion cause accurately represents the root cause of the emotion. For tweets that describe multiple events, the principal source of emotion should be the targeted event for the emotion cause. **Fluency** looks at the grammatical correctness of the response. To ensure **consistency** between the tweet

and its summary, evaluators check that the details align. They verify the emotion label's accuracy and evaluate the emotion cause on a scale of 1 to 5.

We compared data quality from human annotators and our model. Specifically, we want to understand the differences in the interpretation of emotions and their causes in tweets. To achieve this, we asked annotators to identify the emotion and describe its cause. We provided four potential emotion label options based on the tweet's emotion words. If there were fewer than four emotion words, we selected labels with similar affective properties. We used AffectiveSpace (Cambria et al., 2015) to identify emotion words with similar affective properties and provided four pre-generated options for emotion labels. Annotators could provide their own response if none of the options represented the expressed emotion. For the cause description, we offered pre-generated summaries and allowed annotators to supply their own response.

### 4.2 Evaluation procedure

We sourced annotators from Amazon Mechanical Turk, with selection criteria including being a native English speaker and having completed over 500 HITs with a 95% or above acceptance rate. We conducted a qualification task involving annotating 10 tweets and inspected the quality manually before inviting qualified workers to rate our data.

To validate our dataset, we randomly selected 300 samples and had three annotators answer four questions about the emotion and summary of the cause. Annotators were paid an average of $12 per hour. Appendix B contains more information about our Mechanical Turk task.

### 4.3 Evaluation Results

| METRIC | *Relevance* | *Fluency* | *Coherence* |
|---|---|---|---|
| AVERAGE | 4.50 | 4.47 | 4.54 |

Table 3: Emotion cause summarization evaluation score

Table 3 show the human evaluation results. The

average score on **Relevance**, **Fluency** and **Consistency** from human annotators on 300 samples are 4.50, 4.47 and 4.54 respectively. Also, 90% of the emotion causes in our dataset have average rating on all three criteria of 4.0 or above. This shows that the emotion cause generated are of good quality, where the abstractive causes are consistent with the emotion and the cause event mentioned in the tweets. Moreover, annotators agree with our emotion label 89.5% of the time, and when asked how would they describe the cause of the emotion, 98% of the annotators chose to use one of the causes provided by the model. This showed that our dataset contains high-quality emotion causes and accurate emotion labels.

## 5 Conclusion

We created a novel, comprehensive dataset of 700K datum points, which aims to simultaneously identify emotions in text and determine the underlying causes behind each emotion. We detailed the data curation, label generation and evaluation pipeline so that researchers can use to create their own datasets that cater their needs. Finally, we published the dataset so that other researchers can make use of the dataset for their tasks.

## 6 Limitations

We believe that our dataset will bring great benefits to the community, however, it is not without limitations. First, the number 48 was chosen heuristically as we explored the dataset. Second, we labeled emotion classes with patterns matching. This is not 100% accurate and cannot detect sarcasm or negations. Emotion and emotion-cause pairs will not be exhaustively extracted in tweets with multiple emotions and multiple causes. Third, we cannot guarantee that tweets with vulgar languages are completely cleaned. Lastly, although the human rating of the cause are very high, the quality of the abstractive cause depends largely on the capability of the pre-trained model.

## 7 Ethical considerations

This research is approved for Internal Board Review exemption, reference code number NUS-IRB-2023-485

## 8 Acknowledgements

This research is supported by SINGA Award for Mia Huong Nguyen and the start-up grant from the Department Information Systems and Analytics, School of Computing, NUS for Suranga Nanayakkara.

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

| Search phrase | num tweets |
|---|---|
| "("I feel sad" OR "I feel happy" OR .. OR "I feel drained")" | 1,300,928 |
| "("I'm sad" OR "I'm happy" OR .. OR "I'm drained")" | 3,109,289 |
| "("Im feeling" OR "I am feeling" OR "I'm feeling" OR "I feel") because" | 3,798,830 |
| "("I'm feeling sad because" OR .. OR ("I'm feeling drained because")" | 230,191 |
| "("I'm feeling sad" OR "I'm feeling happy" OR .. OR "I'm feeling drained")" because | 120,218 |
| "("I'm sad because" OR "I'm happy because" OR .. OR "I'm drained because")" | 408,291 |
| "("Im" OR "I am" OR "I'm" OR "iam" OR "i am" OR "i'm") (happy OR... OR drained) because" | 930,900 |

Table 4: Search phrases and their corresponding number of tweets

## A Data Curation Pipeline

Data was curated by scraping tweets from Twitter using the Twitter API over a period of 15 years, spanning from 2008 to 2022. To extract tweets containing emotional expressions and potential emotion causes, we refined search phrases iteratively. We narrowed down the search by searching for tweets where users explicitly express their emotions and the cause of their emotions using the following search phrase: "("Im feeling" OR "I am feeling" OR "I'm feeling"- like) because" This search phrase yields 350,000 tweets. By following prior works, such as Mohammad et al. (2014) and Sosea et al. (2022), we cleaned the tweets by removing emojis, emoticons, hashtags, and duplicated letters. The data that had been cleaned was then processed using a heuristic algorithm based on predefined rules. This algorithm aimed to extract emotions and the causes of those emotions. The cleaned tweets were divided into individual tokens and tagged with their respective parts of speech using the Part-of-Speech (POS)[3]. tagging technique. In each tweet, we searched for the phrases "I feel" or "I am feeling" and analyzed the six words that followed those phrases to identify words related to emotions. We assigned the emotion word found as the label for the emotion in that tweet. Typically, emotion-case datasets are limited to only six to eight labels. However, upon analyzing the initial results, we discovered a broad spectrum of words associated with emotions. To address this, we examined the frequency of occurrence for different emotion words and identified the top 48 most frequently used ones. This list served as a tool to enhance the search process by reducing noise and focusing on relevant data points. The updated search phrases were specifically designed to iden-

tify tweets that included phrases like "I am," "I feel," "I am feeling," and so on, in combination with the list of emotions we had previously identified. In total, we tested eight different search phrases, resulting in an initial dataset of 9.8 million tweets. The table 4 provides additional information about each search phrase and the corresponding number of curated tweets.

To ensure the desired emotion pattern ("I am/feel/am feeling ..."), we applied additional filtering rules. We excluded tweets where emotions were not structured in a meaningful order, like "I am," "emotionX," and "because" within the same tweet. Phrases like "I feel bad for ..." and "I feel sorry for ..." expressed sympathy rather than the emotion "bad," so we marked them as sympathy and excluded them from the final dataset. Additionally, we filtered out offensive or inappropriate content by comparing the tweets against a list of offensive words[4].

To further refine the dataset, we excluded tweets where no phrases appeared after "because" since they were less likely to express the cause of an emotion. As a result of this filtering process, the dataset was reduced to around 3.5 million tweets. To enhance the dataset's quality, we selected data points that adhered to specific patterns, such as

- "I X $(e_1|e_2|..|e_{50})$ because"
- "I X (w+) $(e_1|e_2|..|e_{50})$ because" where X in {am, feel, am feeling}

where $e_i$ represents the emotion word; (w+) is a place holder for exaggerators such as "very", "extremely", etc. These patterns are the most explicit and direct patterns that people can use to express their emotions and the cause behind them. After applying the filter, the dataset was reduced to 800,000 tweets.

---

[3]https://www.nltk.org/

[4]https://www.cs.cmu.edu/ biglou/resources/bad-words.txt

# B Human Evaluation

I feel guilty because I miss classes to listen to a colloquium but that's stupid because I'm supposed to have a project and a research and do my best for it but I still feel bad

**Question 1: What is the emotion expressed in the text? Use the original word where applicable.**

Example: Tweet: Everyday of my life I feel grateful because I am so blessed with not one but 2 mothers who love. Emotion: grateful

- guilty
- stupid
- bad
- low
- None of above: _______________

I feel guilty because I miss classes to listen to a colloquium but that's stupid because I'm supposed to have a project and a research and do my best for it but I still feel bad

**Question 2: How would YOU describe the cause behind the expressed emotions?**

Example: Tweet: Everyday of my life I feel grateful because I am so blessed with not one but 2 mothers who love. Cause: having 2 loving moms

- Some of the:
- missed classes for colloquium instead of project/research
- missing classes to listen to a colloquium instead of working on projects
- missing classes to listen to a colloquium
- missed classes to listen to colloquium
- None of the above: _______________

Figure 3: Human Evaluator Questions

Tweet: I feel guilty because I miss classes to listen to a colloquium but that's stupid because I'm supposed to have a project and a research and do my best for it but I still feel bad
Emotion: guilty
Cause: missed classes for colloquium instead of project/research

**Question 3: Does the emotion label accurately reflect the user's expressed emotion?**

- Yes
- No

Figure 4: Human Evaluator Questions

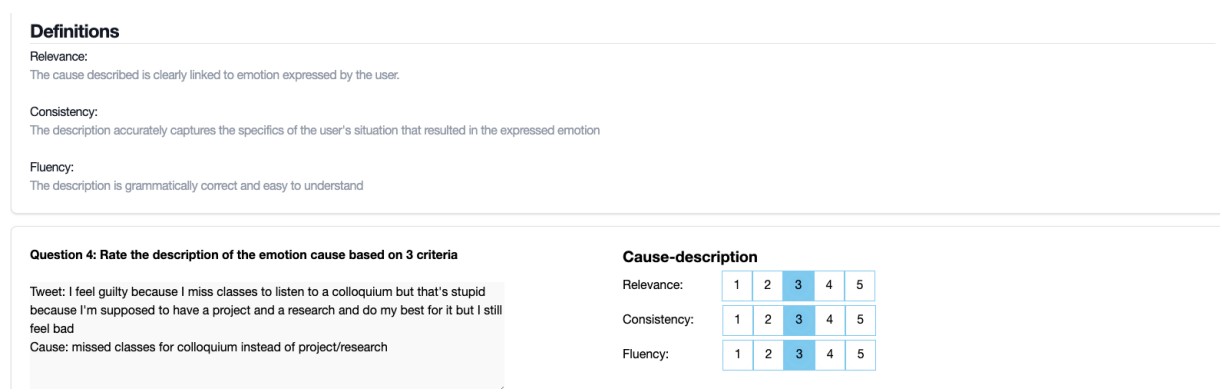

**Definitions**

Relevance:
The cause described is clearly linked to emotion expressed by the user.

Consistency:
The description accurately captures the specifics of the user's situation that resulted in the expressed emotion

Fluency:
The description is grammatically correct and easy to understand

Figure 5: Human Evaluator Questions

# C Data Distribution

| Emotion | #tweet | Percentage | Emotion | #tweet | Percentage |
|---|---|---|---|---|---|
| bad | 166732 | 23.86% | nervous | 11515 | 1.65% |
| happy | 126106 | 18.05% | good | 11231 | 1.61% |
| excited | 40010 | 5.73% | scared | 10923 | 1.56% |
| upset | 37276 | 5.33% | proud | 9356 | 1.34% |
| sad | 31574 | 4.52% | grateful | 6978 | 1.00% |
| sorry | 23864 | 3.41% | frustrated | 6405 | 0.92% |
| tired | 19965 | 2.86% | terrible | 6192 | 0.89% |
| depressed | 19434 | 2.78% | emotional | 6023 | 0.86% |
| sick | 17644 | 2.52% | confident | 4944 | 0.71% |
| guilty | 17440 | 2.50% | anxious | 4820 | 0.69% |
| lonely | 15558 | 2.23% | great | 4739 | 0.68% |
| blessed | 15104 | 2.16% | awful | 4308 | 0.62% |
| angry | 14274 | 2.04% | lost | 4175 | 0.60% |
| stupid | 13813 | 1.98% | down | 3473 | 0.50% |
| lucky | 12650 | 1.81% | uncomfortable | 3146 | 0.45% |
| stressed | 11836 | 1.69% | nauseous | 2474 | 0.35% |
| insecure | 2265 | 0.33% | exhausted | 1920 | 0.28% |
| helpless | 1831 | 0.27% | overwhelmed | 1632 | 0.24% |
| hopeful | 696 | 0.10% | optimistic | 875 | 0.13% |
| motivated | 634 | 0.09% | low | 543 | 0.08% |
| inspired | 542 | 0.08% | nostalgic | 541 | 0.08% |
| hopeless | 510 | 0.07% | drained | 495 | 0.07% |
| positive | 449 | 0.07% | discouraged | 389 | 0.06% |

Table 5: Distribution of Emotion Classes