# OpenReview forum: "EMO-KNOW: A Large Scale Dataset on Emotion-Cause"
_EMNLP/2023/Conference — EMNLP 2023 Findings_

### Official Review · Reviewer_NfoM · 2023-07-31

**Typos Grammar Style And Presentation Improvements:** See reasons to reject.
**Soundness:** 3

**Ethical Concerns:**

Yes

**Excitement:**

3: Ambivalent: It has merits (e.g., it reports state-of-the-art results, the idea is nice), but there are key weaknesses (e.g., it describes incremental work), and it can significantly benefit from another round of revision. However, I won't object to accepting it if my co-reviewers champion it.

**Justification For Ethical Concerns:**

How did the authors handle the privacy issues of tweets carrying personal information?

**Missing References:**

None.

**Paper Topic And Main Contributions:**

The paper presents a new large-scale emotion-cause dataset with over 700,000 tweets and 48 emotion classes. The main contributions are the extensive number of data instances contained and the range of emotion classes enabling subtle differences between emotions within the same category.

**Questions For The Authors:**

See reasons to reject.

**Reasons To Accept:**

1. Large-scale dataset containing 700,000 tweets, each with
2. 48 emotion categories, which are more refined than the classic 7 emotion categories mostly used in existing datasets, enabling more refined emotion-cause analysis.

**Reasons To Reject:**

1. Not strong or clear motivation: While I mainly agree that this new dataset is beneficial, its motivation is not clear. The authors claimed that coarse granularity is not good enough and finer granularity is needed but didn't explicitly analyze how finer granularity is beneficial. Take "pride" and "gratitude" as examples, why is knowing the causes of these similar but still different categories important? What would without knowing this cause? On the other hand, the authors also claim that abstractive emotions are desired and better, but why? How are they better and more beneficial? Stronger motivations are expected.

2. Not clear description: how did the authors end up with 48 categories? I cannot find any relevant description in the manuscript. If this information is contained in the appendix, then this should be added to the manuscript since the guideline states explicitly that the manuscript is a standalone document with every information needed to understand the paper.

3. The given limitations are not relevant enough: the authors haven't claimed to build a balanced dataset, so the balance problem is not the main limitation in this paper; it is a challenge that should be solved by other researchers if they are using the dataset. In my opinion, the limitation should be analyzed from the determination of the 48 categories, and the process of the data cleaning and labeling process. For example, how can the authors guarantee that 48 categories are enough, or there are no circumstances that sometimes two categories have the same meaning and mostly they are similar but different? More insightful limitation analyses are expected.

4. Somehow poor readability:
- confusing claims and reference: the authors cited (Barrett, 2017) that concluded emotions are not simply reducible to a basic set of broad categories, but what defines "broad" and "basic"? The authors use 48 emotion categories. Are these 48 categories not broad and basic? What makes them different from the existing categories and why?
- repetitive description, such as lines 054-057 and lines 066-068;
- confusing descriptions:
  - In lines 157-160, what are the "outlined structure" and "this pattern"? Am I missing something or is it put in the appendix? If it's the latter, same problem as the second comment
  - In line 162, "most cases follow this rule", exactly what is the ratio here? Please put all necessary information in the manuscript for better readability.

**Reproducibility:**

3: Could reproduce the results with some difficulty. The settings of parameters are underspecified or subjectively determined; the training/evaluation data are not widely available.

**Reviewer Confidence:**

4: Quite sure. I tried to check the important points carefully. It's unlikely, though conceivable, that I missed something that should affect my ratings.

---

> ### Author Rebuttal · Authors · 2023-08-29
>
> Thank you for your thoughtful comments. We understand your concerns and would like to address them as follows:
> 1. About the motivation of building the dataset:
> We grounded our work in the most recent advances in affective science. Although researchers still debate about the nature of emotions, whether emotions are basic and universal, and how many basic emotions are there, most have agreed that there are a lot more than 6 basic emotions, as laid out in Ekman’s model [1, 2, 3]. Most datasets in emotion cause analysis are built on this outdated view of emotion, hindering scientists to build models that predict emotion and emotion causes more accurately.
> Consider a scenario:
> *"Tim and Henry both receive recognition at work for their exceptional contributions to a project. Tim feels a strong sense of pride, while Henry feels immense gratitude for the recognition. Earlier, Tim had work his heart out, spent days and nights overcoming so many difficulties to achieve his goal; while Henry felt that he was lucky and he got the help from other teammates."*
> Without knowing the causes behind their emotions, it might be easy to assume that both are simply experiencing positive emotions due to the recognition they received.
> Our dataset is not only large in size, but it has 2 important  features:
>    - It is rich in its classes (48 emotion categories),
>    - It provides an abstractive emotion cause (not abstractive emotion). The abstraction of emotion cause allows generalization across contexts (e.g. being more fortunate than other people is a common cause for gratitude, but it can be expressed differently in different contexts). This is shown in Table 2. We will elaborate more on this in the revised paper.
>
> The motivation of building a dataset with abstractive emotion cause is to facilitate emotional reasoning, which is crucial for building models that have a deep understanding of human emotions and do not make stereotypical assumptions about emotions.
>
>  We agree that the second paragraph of our introduction is duplicated with poor readability, while the first paragraph has not motivated well enough the need to have a large dataset with a wide range of emotion classes and abstractive causes. We will update our manuscript to address these concerns.
>
> 2. About the number 48 emotion classes:
> Thank you for raising this up. We agree that we should have included the information in the main manuscript.
> We mostly ground our work in the Theory of Constructed Emotions, where one’s emotion “nauseous'' can be totally different from their emotion “sick” even when others think they are the same. Following the recommendation by Barrett & Westlin 2021, section 2.4.2, para. 3 [4], to preserve the ecological accuracy of the dataset, we keep the emotion words used by people as they are, and do not follow prior works to group them into more “basic” categories (like 28 categories in GoEmotion[5] or 32 in EmpatheticDialogue[6]). This is a strength of the dataset, as it is ecologically accurate, but can make it difficult to combine with other datasets. We briefly discussed our exploration of emotions expressed in Twitter in Appendix A. We examined the frequency of emotion words on a sample of 300K tweets and picked a cut-off at 48. We will update this information in our revised manuscript.
> 3. About Limitations:
> We largely agree with the reviewer that the limitation section should focus more on the method that we used to construct the dataset. First, the number 48 was chosen heuristically as we explored the dataset. Second, we labeled emotion classes with patterns matching. This is not 100% accurate and cannot detect sarcasm or negations. Emotion and emotion-cause pairs will not be exhaustively extracted in tweets with multiple emotions and multiple causes. Third, we cannot guarantee that tweets with vulgar languages are completely cleaned. Lastly, although the human rating of the cause are very high, the quality of the abstractive cause depends largely on the capability of the pretrained model.
>
> We acknowledge that our limitation sections can be improved and will update this part in our manuscript.
>
>
> 1. Coppini, S., Lucifora, C., Vicario, C.M. et al. Experiments on real-life emotions challenge Ekman's model. Sci Rep 13, 9511 (2023).
> 2. Cowen, Alan S., and Dacher Keltner. "Self-report captures 27 distinct categories of emotion bridged by continuous gradients." Proceedings of the national academy of sciences 114.38 (2017): E7900-E7909.
> 3. Ortony, A. (2022). Are All “Basic Emotions” Emotions? A Problem for the (Basic) Emotions Construct. Perspectives on Psychological Science, 17(1), 41–61
> 4. Barrett, Lisa Feldman, and Christiana Westlin. "Navigating the science of emotion." Emotion measurement. Woodhead Publishing, 2021. 39-84.
> 5. Demszky, Dorottya, et al. "GoEmotions: A Dataset of Fine-Grained Emotions." Proceedings of the 58th Annual Meeting of the Association for Computational Linguistics. 2020.
> 6. Hannah Rashkin, Eric Michael Smith, Margaret Li, and Y-Lan Boureau. 2019. Towards Empathetic Open-domain Conversation Models: A New Benchmark and Dataset. In Proceedings of the 57th Annual Meeting of the Association for Computational Linguistics, pages 5370–5381, Florence, Italy. Association for Computational Linguistics.

---

### Official Review · Reviewer_mpV3 · 2023-08-02

**Soundness:** 2

**Excitement:**

3: Ambivalent: It has merits (e.g., it reports state-of-the-art results, the idea is nice), but there are key weaknesses (e.g., it describes incremental work), and it can significantly benefit from another round of revision. However, I won't object to accepting it if my co-reviewers champion it.

**Paper Topic And Main Contributions:**

The paper introduces a large-scale dataset of emotion causes, and describes the pipline of dataset construction. However, author(s) did not  indicate the motivation of constructing this dataset and the advantages of the proposed dataset compared with the existing emotion cause dataset.

**Questions For The Authors:**

1.  will the finer granularity of emotion categories bring benefits to the tasks of cause detection and emotion-cause pair detection?
2.  lines 206-207 mentioned that author(s) adopted prompt to obtain emotion cause label from GPT3.5. So, what is the prompt used in this paper like? can author(s) provide a prompt template used?

**Reasons To Accept:**

1. author(s) constructed a large-scale dataset for emotion cause analysis, and it might bring benefits for the further study on the application of pretrained model with the emotion cause analysis.

**Reasons To Reject:**

1. the motivation of constructing finer granularity of emotion categories is not clear. I felt confusing about the profits of finer finer granularity of emotion categories in emotion cause analysis.
2.  the model is required to have strong understanding capacity to predict the finer granularity of emotion categories, which means the model is more possible to make wrong prediction. Further, the model will face bigger challenge based on a wrong emotion prediction.

**Reproducibility:**

3: Could reproduce the results with some difficulty. The settings of parameters are underspecified or subjectively determined; the training/evaluation data are not widely available.

**Reviewer Confidence:**

4: Quite sure. I tried to check the important points carefully. It's unlikely, though conceivable, that I missed something that should affect my ratings.

---

> ### Author Rebuttal · Authors · 2023-08-28
>
> Thank you for your comments. We appreciate your time and we understand the concerns that you raised. We would like to address them as follows:
> 1. We agree that our introduction has not made it clear the need for having a dataset with rich emotion categories. Prior work largely agree that human emotions are complex, multifaceted, and not reducible to a set of 6 basic emotions (happy, sad, angry, surprise, disgust, fear) as in Ekman’s model[1]. Having finer granularity of emotion labels means richer information of emotion and the context that give rise to such emotions. A model that can distinguish between pride and gratitude would be able to give a more tailored response/prediction/reasoning about the emotional information it is processing than a model that can only distinguish happy and sadness. This is precisely the reason why we see a growing number of datasets that go beyond 6-7 emotions classes.[ 2, 3]
>
> 2. We used pattern matching (described in appendix A) and mentioned in line 160, selection 3.2.1 for the precise reason: avoid the need to build a model that classifies tweets in 48 emotion categories. Our models do not predict emotion categories, we select tweets where emotion labels are mostly obvious to extract to ensure the quality of our dataset. This is reflected in our evaluation section.
>
> As for your questions, please find the answers in the following:
> 1. The finer granularity will bring more nuanced prediction results for models in emotion cause detection and emotion-cause pair prediction.
>  Let’s take an example:
> 	‘This morning I came to work late. There was a car accident so my bus couldn’t move. I got stuck in traffic for 1 hour. When I came to the office, my colleague told me his friend got into an accident and was in a critical condition. I feel sorry for the guy, and I am thankful that death has spared me”.
> Without a “grateful” category, it is easy to classify this text as sadness and predict the cause for the emotion as “came to work late”. Doing so would completely miss the point of the text.
>
> 2. . Thank you for point this out. The prompt is: "Answer the following questions: 1. What is the emotion expressed in the text? 2. What is the cause of such emotion? Provide the answer in the following format strictly: 1. E where E is the emotion from this list: [{list of 48 emotions}], 2. C where C is the summary of the cause in 10 words. Use NA for E when there is no emotion, and use NA for C when there is no cause.”
> We will revise our manuscript and add this to the main pages.
>
> **References**:
> 1.  Paul Ekman. 1992b. An argument for basic Emotion. Cognition & Emotion, 6(3-4):169–200.
> 2.  Demszky, Dorottya, et al. "GoEmotions: A Dataset of Fine-Grained Emotions." Proceedings of the 58th Annual Meeting of the Association for Computational Linguistics. 2020.
> 3.  Hannah Rashkin, Eric Michael Smith, Margaret Li, and Y-Lan Boureau. 2019. Towards Empathetic Open-domain Conversation Models: A New Benchmark and Dataset. In Proceedings of the 57th Annual Meeting of the Association for Computational Linguistics, pages 5370–5381, Florence, Italy. Association for Computational Linguistics.

---

### Official Review · Reviewer_XtNV · 2023-08-02

**Soundness:** 2

**Excitement:**

2: Mediocre: This paper makes marginal contributions (vs non-contemporaneous work), so I would rather not see it in the conference.

**Paper Topic And Main Contributions:**

This paper describes a twitter data scraping pipeline for emotion and emotion cause related tweets collection and the use of SOTA LLMs to find out the emotion cause of a tweet provided an emotion by formulating the task as a prompt to the LLMs as a question; "provided an emotion e, what is the cause of it?"

**Questions For The Authors:**

- Why sympathy is not counted as another emotion?
- Why only emotions are extracted from tweets having specific text patterns, how is it novel from
other works?
- How the bias is not introduced the way you have scraped data.

**Reasons To Accept:**

- Large scale data collection approach
- Human annotation efforts

**Reasons To Reject:**

- Emotion cause extraction is done on a sample which is very explicit on containing emotion and their cause. This is because the authors used a particular pattern to extract those.
- No novelty on data modelling, use of existing models to extract emotion through formulating it to text summarization task
- Number of emotions identified is not properly justified.
- No analysis on hard examples where there is overlap in emotions but causes separates them.

**Reproducibility:**

1: Could not reproduce the results here no matter how hard they tried.

**Reviewer Confidence:**

4: Quite sure. I tried to check the important points carefully. It's unlikely, though conceivable, that I missed something that should affect my ratings.

---

> ### Author Rebuttal · Authors · 2023-08-28
>
> Thank you for your valuable feedback. We would like to address your concerns as follows:
> 1. Emotion cause extraction done on explicit sample containing emotion and their cause
>  - We do not claim a novel method to predict emotions.
>  - The use of patterns to extract users' emotion is to capture the authentic emotion of users, in their own words, in contrast with hiring human annotators to annotate what the annotator *think* the users’ emotions to be. **This is to warrant the quality of our dataset**.
>  - Emotions are very personal and subjective experiences where expressions of emotions vary among individuals and can be significantly different across cultures[1, 2,3]. What human annotators can provide is their best guesses of users’ emotions, which are often subject to social bias and stereotypes.
>  - Though using pattern matching to extract emotion labels appears to be simplistic, this approach provides an effective and elegant solution to the issue of bias in emotion annotations.
>
> 2. No novelty on data modelling
> - We believe we should have better highlighted the main contribution of the paper (which we will address in the revised version). Our main contribution is a large-scale dataset of **rich emotion classes** and **abstractive emotion causes**, serving as a knowledge base that enables other researchers to build models that understand the nature of human emotions better.
> - By abstractive emotion cause, we mean a generalizable cause that can be adapted to different situations. Consider these two examples:
>   - “If you are not making mistakes, you’re not doing anything. I feel positive because I make mistakes and I can be better after that.
> Emotion: positive.
> Cause: overcoming mistakes and becoming a better person.
> And:
>   - 'When setbacks arise, I see them as opportunities for growth. I feel motivated because each mistake I make is a chance to learn and improve."
>   Emotion: motivated.
>   Cause: overcoming mistakes and becoming a better person.
>
> In the two cases, the abstractive cause is the same but expressed in different wordings.
> - Having abstractive emotion-cause enables AI models to learn underlying patterns, enhancing the ability to predict emotional responses across a wide array of contexts. This is the novel contribution of our approach.  In contrast, prior work mainly used extensive annotation to extract parts of the text that contains emotion-cause, making the extracted cause very context-specific and hard to generalize.
>
> 3. Number of emotions identified is not properly justified.
> Thank you for pointing this out. We briefly discussed our exploration of emotions expressed in Twitter in Appendix A (line 516-518). We examined the frequency of emotion words on a sample of 300K tweets and picked a cut-off at 48. We will move this info from the appendix to the main body of the paper n in our revised manuscript.
>
> 4. No analysis on hard examples where there is overlap in emotions but causes separates them.
> Thank you for your comment. Table 2 provides a glance of how emotions of related categories can have very different causes. For example, “proud” and “inspired” will be categorized as happiness in all other datasets, but they have very different sets of causes. Given the page limit, we had to prioritize other information to show you that the dataset that we constructed is of high quality and is rigorously validated. We will add more details explaining Table 2.
>
> ### For the questions:
> 1. Why is sympathy not counted as another emotion?
> We eliminated “expression of sympathy” for tweets with the pattern “I feel bad/sorry for …” since the person is not really feeling “bad”, and the clause following “for” often describes the situation of someone else and not the users. For the purpose of capturing users’ true emotions and emotion cause, we eliminated these expressions to ensure the quality of our dataset.
> 2. Why only emotions are extracted from tweets having specific text patterns, how is it novel from other works?
> As stated in point 2, the purpose of patterns is to eliminate the need to manually annotate emotion labels (which is costly and not scalable), overcoming the problem of social bias and stereotypes. Other datasets assumed emotions are of only 6-8 categories, overlooking the nuance differences between fine-grained emotions such as proud, gratitude, inspired, lonely, nostalgic, etc. Without patterns, these fine-grained emotion labels need to be manually labeled by annotators - a process that is very costly and prone to biases.
> 3. How the bias is not introduced the way you have scraped data?
> There are several ways that bias can be introduced in the way we scraped the data:
> Tweets with emotional expressions that do not follow our defined patterns will not be selected into the dataset.
> Tweets with emotion words that are not in our list will also be absent.
>             Although these selection biases exist, it does not affect the quality of our dataset, that is the size of the dataset, the richness of emotion label, the generalizability of emotion causes.
>
> **References**:
> 1. Hendrik Schuff, Jeremy Barnes, Julian Mohme, Sebastian Padó, and Roman Klinger. 2017. Annotation, Modelling and Analysis of Fine-Grained Emotions on a Stance and Sentiment Detection Corpus. In Proceedings of the 8th Workshop on Computational Approaches to Subjectivity, Sentiment and Social Media Analysis, pages 13–23, Copenhagen, Denmark. Association for Computational Linguistics.
> 2. Troiano, E., Padó, S., & Klinger, R. (2021). Emotion Ratings: How Intensity, Annotation Confidence and Agreements are Entangled. Workshop on Computational Approaches to Subjectivity, Sentiment and Social Media Analysis.
> 3. Sven Buechel and Udo Hahn. 2017. Readers vs. Writers vs. Texts: Coping with Different Perspectives of Text Understanding in Emotion Annotation. In Proceedings of the 11th Linguistic Annotation Workshop, pages 1–12, Valencia, Spain. Association for Computational Linguistics.

---

### Meta-Review · Area_Chair_Gh9g · 2023-09-17

**Recommendation:** 3

**Metareview:**

The paper presents a method for collecting Twitter data related to emotions and their causes, resulting in the creation of a large-scale dataset. This dataset, sourced from tweets, is designed to identify the cause of a given emotion by prompting state-of-the-art Language Models (LLMs) with specific questions. The dataset encompasses over 700,000 tweets and is categorized into 48 distinct emotion classes. These classes aim to capture nuanced differences between emotions that may fall under a broader category. However, the paper does not clearly outline the motivation behind the creation of this dataset or how it compares to existing datasets in terms of advantages.

The paper's strengths lie in its ambitious data collection approach, with a significant emphasis on human annotation. The creation of a dataset with 700,000 tweets and 48 emotion categories is commendable, as it offers a more detailed perspective on emotions compared to existing datasets. This refined approach to categorizing emotions could be beneficial for in-depth emotion-cause analysis and might aid in the application of pretrained models in this area.

The method used for emotion cause extraction seems to be based on a specific pattern, which might not capture the full spectrum of emotions in real-world scenarios. The paper's approach lacks novelty, especially in data modeling, and there's an over-reliance on existing models. The authors, during the rebuttal phase, convincely answered some concers of the reviewers.

---

### Decision · Program_Chairs · 2023-10-07

**Decision:**

Accept-Findings

**Comment:**

The paper presents a method for collecting Twitter data related to emotions and their causes, resulting in the creation of a large-scale dataset. This dataset, sourced from tweets, is designed to identify the cause of a given emotion by prompting state-of-the-art Language Models (LLMs) with specific questions. The dataset encompasses over 700,000 tweets and is categorized into 48 distinct emotion classes. These classes aim to capture nuanced differences between emotions that may fall under a broader category. However, the paper does not clearly outline the motivation behind the creation of this dataset or how it compares to existing datasets in terms of advantages.

The paper's strengths lie in its ambitious data collection approach, with a significant emphasis on human annotation. The creation of a dataset with 700,000 tweets and 48 emotion categories is commendable, as it offers a more detailed perspective on emotions compared to existing datasets. This refined approach to categorizing emotions could be beneficial for in-depth emotion-cause analysis and might aid in the application of pretrained models in this area.

The method used for emotion cause extraction seems to be based on a specific pattern, which might not capture the full spectrum of emotions in real-world scenarios. The paper's approach lacks novelty, especially in data modeling, and there's an over-reliance on existing models. The authors, during the rebuttal phase, convincely answered some concers of the reviewers.